# Natural Deep Eutectic Solvents (NADESs) Combined with Sustainable Extraction Techniques: A Review of the Green Chemistry Approach in Food Analysis

**DOI:** 10.3390/foods12010056

**Published:** 2022-12-22

**Authors:** Ciro Cannavacciuolo, Stefania Pagliari, Jessica Frigerio, Chiara Maria Giustra, Massimo Labra, Luca Campone

**Affiliations:** Department of Biotechnology and Biosciences, University of Milano-Bicocca, Piazza Della Scienza 2, I-20126 Milano, Italy

**Keywords:** natural deep eutectic solvents, NADES, green extraction, food analysis

## Abstract

Usual extraction processes for analyzing foods, supplements, and nutraceutical products involve massive amounts of organic solvents contributing to a negative impact on the environment and human health. In recent years, a new class of green solvents called natural deep eutectic solvents (NADES) have been considered a valid alternative to conventional solvents. Compared with conventional organic solvents, NADES have attracted considerable attention since they are sustainable, biodegradable, and non-toxic but also are easy to prepare, and have low production costs. Here we summarize the major aspects of NADEs such as the classification, preparation method physicochemical properties, and toxicity. Moreover, we provide an overview of novel extraction techniques using NADES as potential extractants of bioactive compounds from foods and food by-products, and application of NADEs in food analysis. This review aims to be useful for the further development of NAES and for broadening the knowledge of these new green solvents in order to increase their use for the extraction of bioactive compounds and in food analysis.

## 1. Introduction

The analysis of foods is a comprehensive process of extraction, identification, and quantification of several classes of compounds from natural matrices. The detection and quantification of primary metabolites (sugars, amino acids, vitamins, and lipids), contaminants (toxins, heavy metals, and allergens), and secondary metabolites (polyphenolics, flavonoids, terpenes, and alkaloids) is a crucial practice for ensuring the safety and quality of foods and related functional products. Due to the variable structure of food analytes, a gap in a universal method suitable for the extraction and analysis of all compounds is lacking. Moreover, conventional extractants are usually made of organic solvents and common extraction techniques usually require a long extraction time to exhaust the matrix [1]. The actual discussions about climatic changes provide a growing awareness of the scientific and industrial community to reduce the environmental impact by using sustainable processes. In general, the main principles of “green chemistry” are based on the design of processes aimed to reduce energy consumption and the use of eco-friendly solvents with less toxicity to the environment and human health. In this view, ionic liquids (ILs) and deep eutectic solvents (DES) replaced the use of traditional organic solvents in several analytical applications of the last years.

The development of ionic liquid (ILs) solvents started in the 90s. Despite their nontoxic behavior as pure solvents, several steps of purification should be used to produce ILs that negatively affect economic and environmental sustainability [2]. Thus, deep eutectic solvents (DES) were born due to their non-toxic handling, wide versatility of use in different sectors, and sustainable production techniques that can offer a more environmentally sustainable alternative [3]. Generally, deep eutectic solvents (DES) are liquid mixtures, at room temperature, characterized by a lower eutectic point than pure forms of the constituents [4]. A compound acting as a hydrogen bond donor (HBD) and a compound that acts as a hydrogen bond acceptor (HBA) are the main components of the mixture [5,6] sometimes the addition of a third component, usually water, is needed. The general formula (Cat+ X- zY) describes a typical DES system. Cat+ is usually an ammonium, phosphonium, or sulfonium cation; X- is a Lewis base, commonly a halide anion; Y is a Lewis or Brønsted acid, and z represents the number of Y molecules interacting with the anion X- [7,8].

Since 2014, natural deep eutectic solvents (NADES) have been developing as a special group of DES. These emerging systems are made by the combination of natural molecules, allowing further improvement in the impact on the environmental and toxicity risk compared with other deep eutectic solvents. NADES have been studied since 2011 and a rising number of papers highlighted their applications in several fields of chemistry. Among the reported studies, since 2014, the focus on NADES’ applications, physiochemical properties, and combination with innovative extraction techniques was particularly detailed for their applications in foods analysis (Figure 1).

Generally, the subgroup of NADESs uses HBD and HBA made by sugars, alcohols, organic acids, amino acids, or amines naturally occurring in plant metabolism or eucaryotic cellular systems [9]. NADESs have been indicated as possible solvents present in living cells, thus explaining the presence of compounds at much higher concentrations than what is soluble in aqueous solutions [10]. Furthermore, NADESs reduce physicochemical constraints of metabolite transport and cellular processes through the formation of liquid microenvironments [11]. The mostly used HBA are chlorine chloride (ChCl) and betaine for their low cost, non-toxicity, and biodegradability [8,12] but also proline, glycine, alanine, histidine, lidocaine, acetylcholine chloride, and nicotinic acid are currently used as eco-friendly and biocompatible molecules [13]. Natural carboxylic acids (gallic acid, benzoic acid), hydroxycinnamic acid derivatives (coumaric acid and caffeic acid) different sugars (xylitol, glucose, fructose), organic acids (oxalic acid, malic acid), and fatty acids (stearic acid, oleic and linoleic acid) are the used as natural HBD combined with HBA [13].

The advantage of these solvents is related to chemical properties, such as low melting points, low volatility, nonflammability, low vapor pressure, polarity, chemical and thermal stability, and miscibility solubility [5,6,7,14,15]. Additionally, a convenient atom economy and a low environmental impact are related to their low costs and high yields of production. Their assembly through intermolecular hydrogen bond interactions does not involve chemical reaction hence there is no production of secondary compounds reducing the need for further purification steps, and no waste is usually produced [7,15,16,17]. The current review is focused on the topic of natural deep eutectic solvents (NADES) combined with different extraction techniques to design processes that reduce energy consumption, ensure the use of safe alternative solvents, and optimize high extraction efficiency of target molecules occurring in several fields of foods analysis.

## 2. Physicochemical Properties of NADES for the Extraction Process

The intermolecular interactions, mainly hydrogen bond interactions, between the HBA and the HBD are responsible for the physicochemical properties of the DES [14,18]. By changing either the HBA/HBD ratio or one of the components, the NADES can be specifically tailored for different applications. This is a major advantage in terms of achieving desirable properties and improving extraction efficiency [7,17]. Viscosity, polarity, density, and pH condition are the main physicochemical properties affecting the extraction of natural compounds from food matrices. The addition of water as a third component of the system can modulate the conditions of the NADESs for the extraction of several compounds and chemicals for food analysis applications.

Generally, the high viscosity of HDA and HDB mixtures cause a low mass transfer phenomenon affecting the extraction efficiency. Temperature can drastically reduce the viscosity but a mediation with the thermolability of natural compounds must be considered. Added water, reported in a range of 10% to 80% [19], disrupts the hydrogen bond interaction between HAD and HDB modulating the viscosity of the system. Data reported by Zhekenov et al., 2019 suggest a maximum limit of 50% molar fraction of water beyond the system act as a solution [20].

Polarity is the key property to solubilizing metabolites in a solid-liquid extraction. Fixing the choline chloride (ChCl) as HBA, the use of different HBD can affect the polarity of the system. Craveiro et al., 2016 reported the use of sugars (glucose, sucrose, xylose) and organic acids (citric acid, tartaric acid) generating different polarity systems. In particular, NADES composed of ChCl and organic acids resulted more polar than those combined with sugars [21]. The water addition can also affect the polarity of the NADES [22,23]. HBD used for the preparation of NADES act both as hydrophilic and lipophilic components. Components with electronegative groups can form dipole−dipole interactions with polar solvents explaining the hydrophilic properties [24]. On the other hand, natural lipophilic compounds commonly used in NADES are characterized by a polar moiety forming dipole-dipole interactions with polar solvents and a hydrophobic moiety with a tendency to aggregate in aqueous solution to minimize the area of contact between water and nonpolar molecules. Several NADES systems are described by the hydrophilicity/lipophilicity balance phenomenon. For instance, fatty acid- and terpenes-based natural compounds acting as HBD have been reported for NADES [25,26,27].

Complete profiling of complex food matrices by using organic solvents allows the extraction of hydrophilic components or lipophilic components separately while the ignored components that remain in the extraction residues are discarded from the analysis. The advantage to use hydrophilic or hydrophobic NADES is the efficient enhancement of bioactive components with various polarities. The density of hydrophilic or hydrophobic solvents can change allowing a double-phase separation. The two-phase system formed with hydrophilic NADES and hydrophobic NADES can resolve the problem of extraction selectivity and could simultaneously extract the bioactive compounds with various polarities from plant materials. In 2015, Van Osch et al. developed a NADES two-phase system and evaluated the recovery of volatile fatty acids from dilute aqueous solutions [26]. A two-phase aqueous system (NADES–salt solution) has also been established to extract proteins [28,29,30,31] and non-polar anthraquinones [30]. Moreover, Jun Chao et al., 2018 developed a two-phase NADES system (ChCl-LA1/Ch-M/MCO) to extract bioactive metabolites with different polarities from Ginkgo biloba leaves [16].

## 3. Preparation of NADES

The NADESs can be produced from different natural compounds such as ChCl, sugars, amino acids, and polyols. These compounds are in solid form and only after mixing, in a specific combination and molar rates, do they turn into a liquid state [32]. Five methods are reported to prepare NADES: thermal mixing, vacuum evaporation, freeze-drying, ultrasonication, and microwave.

Thermal mixing is a simpler and faster approach that involves mixing and stirring at a temperature of 50–80 °C of two components with water until a colorless liquid is formed [22,33,34].Vacuum evaporation is also similar; the components are dissolved in water and evaporated at about 50 °C in rotary evaporation, and finally, the resulting liquid is placed in a desiccator to reach a constant weight [22,34].Freeze-drying involves dissolving the components in water, then freezing and drying them, resulting in a viscous, transparent solution [34,35].Microwave-assisted synthesis exploits the production of microwaves that upon interaction with precursors generate collisions between molecules and between the hydrogen bond donor and hydrogen bond acceptor components due to dipole rotation resulting in dielectric heating that speeds up the synthesis time [36,37]. According to Popovic et al., 2022, microwaves could be one of the fastest methods for the preparation of some NADES, taking even less than a minute [33]. However, because of the possible overheating caused by the technique, it is advisable to divide the process into several cycles of a few seconds interspersed with cooling pauses [38]. The entire preparation is carried out in closed systems with controlled pressure and temperature.Ultrasound-assisted is a little-explored but effective way of preparing NADESs. The cavitation process promotes, through the release of heat and pressure exerted because of bubbles implosion, the interaction between the hydrogen bond acceptor (HBA) and the hydrogen bond donor (HBD) [34]. According to when described by Santana et al., 2019, the preparation of NADES by ultrasound can also be performed by heating the mixture around 50 °C [34]. This approach requires several minutes with intermediate times between microwaves and the remaining techniques described above.

The preparation of NADES through the different methods (Figure 2) generates clear solvents without the presence of precipitates at room temperature with the same properties [33,34]. The principal difference consists in the timing of the process. Ultrasound and microwave methods offer high speed and efficiency of preparation compared with thermal mixing, vacuum evaporation, and freeze-drying; however, the mixing and stirring method is still widely used because it is very simple, easy to perform and allows the production of high volumes of solvent.

## 4. Use of NADES as Green Solvent in the Extraction Techniques

The use of NADES as extraction solvent is often combined with techniques such as ultrasound-assisted liquid extraction (USAE), microwave (MAE), or pressurized liquid extraction (PLE). This is mainly due to the properties of NADES used as alternative solvents to classical solvents for lower toxicity and higher extraction efficiency, making them suitable for green chemistry extraction techniques that aim to reduce cost, risk, extraction time and environmental impact (Table 1) [33,39]. Numerous studies employing such solvents on food matrices for the recovery of various bioactive compounds are reported in the literature. Mostly, Bajkacz et al., 2017 and Rashid et al., 2023 optimized a method for recovering phenols respectably from soybean and apple by-products by using different types of NADES and ultrasonic techniques. The results demonstrated how the combination of ultrasonic with NADES solvents can be an innovative method of phenol recovery due in part to the H-interactions formed between NADES and phenols; furthermore, the extraction efficiency was implemented by adding 30% water to the NADES mixture, which from the results proved to be the best condition [2,39]. Still, Loarce et al., 2011 have used NADES as modifiers for PHWE extraction, significantly improving anthocyanin recovery compared to water alone while keeping the process green and sustainable [40]. The process was optimized using water with 30% NADES composed of choline chloride and oxalic acid at a temperature of 60 °C as solvent. Another example is the study conducted by Fan and Li., 2022 that through the coupling of NADES and microwaves, finally these solvents were also used with microwave hydrodistillation for the extraction of essential oil from *A. sinensis*. The combination of NADES with microwaves that promote plant cell rupture compared to conventional methods allows implementation of the recovery of molecules by reducing while maintaining the process with low environmental impact and high efficiency [41].

## 5. Toxicity and Sustainability

The green chemistry approach requires minimizing the risks and hazards associated with the process and its impact on the environment and human health. It is precisely for this motive that numerous scholars have begun to move toward the use of safer and more environmentally friendly solvents. At this juncture are NADES, which are considered nontoxic, environmentally sustainable, and biodegradable, thanks mainly to the fact that they are made up of compounds of natural origin. Choline chloride (ChCl), for example, one of the main HBAs, is widely used for the preparation of NADES and used commercially on a large scale as an additive for chicken feed, in addition, NADES are also used for the solubilization of drugs for oral dosing in rats, and the synthesis of biodegradable polyesters with antibacterial properties [42]. The emergence of NADES is related to their very low toxicity, which allows them to be used more safely than the previous ionic solvents (ILs), which, on the other hand, can have similar toxicity to the organic compounds they replace [43].

The toxicity of NADES has long been evaluated by considering only the single toxicity data of all the components used in the preparation, which are reported to be safe [44]. However, more recent studies have indicated that these molecules when mixed may show higher toxicity than the single components, related to the structure of the deep solvents, due to synergistic effects between the individual elements. Hayyan et al., 2013 conducted some experiments to evaluate the toxicity and cytotoxicity of ChCl and phosphonium-based deep solvents [45]. The models used for toxicity evaluation were two strains of Gram+ bacteria *Bacillus subtilis* and *Streptococcus aureus* and two Gram- strains *Escherichia coli* and *Pseudomonas aeruginosa*, while cytotoxicity was considered on *Artemia salina*. The results show that phosphonium-based deep eutectic solvents have higher toxicity on all bacterial strains, highlighting a possible antibacterial effect, unlike choline chloride-based NADES. However, for both types of HBAs, the mixture of NADES was more toxic than the individual components, even for cytotoxicity, the ammonium-based deep eutectic solvents showed a greater effect on *Artemia salina* than the individual components. Justifications for this behavior may be due to several factors such as hydrogen bonding between HBD and the salt anion. It is known that the delocalization of charge that occurs during hydrogen bonding makes the mixture more toxic; in fact, substances with delocalized charges are more toxic than chemicals with localized charges. The lack of oxygen and high viscosity that impairs the movement of *A. salina* may also influence by modifying the toxicity of the mixture. Further experiments conducted by Hayyan et al., 2013 also hypothesize that NADES can interact with cell surfaces and that their accumulation and aggregation may cause increased cytotoxicity [45]. Still, Radosevic et al., 2015 evaluated different aspects of the toxicity of different deep eutectic solvents based on ChCl, in particular, they considered phytotoxicity on wheat, toxicity on fish and human cells, and biodegradability using wastewater microorganisms through closed bottle test [46]. The results show low to moderate cytotoxicity on cells with cell inhibition comparable to that of industrial solvents, inhibition in wheat germination was also not observed with oxidative stress manifestation only at high amount addition, all NADES tested were classified as biodegradable. NADES thus show a good correlation between biodegradability and toxicity with some advantages such as low vapor pressure and low flammability making them preferable to the organic solvents they are supposed to replace. These results, therefore, showed that natural deep eutectic solvents based on choline chloride have a potential green profile and a very good prospect for use; however, considering the different toxicity observed for some mixtures, especially those based on phosphonium, further studies are needed to understand and clear their impact on the environment and organisms [45].

## 6. Recent Application in Food Analysis

### 6.1. Food Analytics

Green chemistry plays an important role in the development of a sustainable process for food manipulation. The optimization of food analysis is a compromise between the reduction of solvents, toxic for human health and environmental pollution, reagents, and energy along with the necessity for high sensitivity, precision, and accuracy for validated analytical methods. NADES have been currently used in green chemistry procedures for sample preparation and analytical workflow focusing on achieving accurate results and prioritizing the sustainability of the processes [47]. In the current review, NADES applications in several fields of food analysis are discussed in procedural workflow steps. Green analytical chemistry plays an important role in the development of a sustainable process for food manipulation. The optimization of sustainable analytical procedures is a compromise between the reduction of solvents, toxic for human health and environmental pollution, reagents, and energy along with the necessity for high sensitivity, precision, and accuracy for validated analytical methods.

The extraction is the core procedure for the recovery of phytochemicals, bioactive compounds from natural sources. The aim to optimize the extraction procedure by choosing solvents, experimental conditions, and extraction techniques is to provide the highest amount of target compounds with minimal contamination of other undesired compounds [48]. Solid–liquid extraction with organic solvents is a conventional procedure used for the recovery of phytochemicals enabling the release of solutes from a solid matrix to the liquid phase usable for further wet analysis as HPLC combined with several UV or MS detectors for compound identification [48]. Nevertheless, pollution enabling, and the toxicity related to the use of these organic solvents is strongly discouraged in the food, cosmetic, and pharmaceutical industries [49]. Natural deep eutectic solvents (NADES) satisfied the environmental consciousness by designing sustainable extractions strategies following the basic principles of green chemistry [50]. Beyond the representation of a sustainable alternative for the extraction of target compounds, these solvents signify an opportunity to develop functional foods or nutraceuticals, using naturally occurring components, colorants, and preservatives [49].

Chromatography is a crucial tool for the analysis of complex food matrices assisting the pre-concentration, separation, and isolation steps in the usual workflows. NADESs are used as stationary phases or eluent in several chromatographic procedures assisting the separation of target compounds from natural products. Tang et al., 2015 used NADES-based stationary phase sorbents. Functionalized spheres were packed in a 250 × 4.6 mm column creating a high-performance size-exclusion chromatography (HP-SEC) system for the separation of metabolites with different shapes and sizes by using water as a mobile phase to separate three polysaccharides alginic acid, fucoidan, and laminarin [13]. A refractive-index detector (RID) was used for testing the resolution of the chromatographic systems. Similarly, Li et al., in 2015 and 2016 used NADES-based silica gel in the HP-SEC column for the separation of dextran molecules of 5, 50, and 670 kDa by using ChCl/B-based and ChCl/AA-based stationary phases showing the best resolution of low-weight dextran (5 and 50 kDa). The separation efficiencies were examined according to their mesoporous structures as pro size and HBD on the surface [51,52]. Tan et al., 2016 observed an improved separation of quaternary alkaloids (coptisine chloride, sanguinarine, berberine chloride, and chelerythrine) on a reverse phase column C-18 consequently the addition of a small amount of deep eutectic solvents in the mobile phase, tested in a concentration range of 0 to 2% *v*/*v*. The improved resolution of the chromatography is explained by the mechanism of binding of both hydrogen-bond donors and hydrogen acceptors combined with the C-18 adsorption of the analytes [53].

Despite the interesting approaches of NADES in analytics, the interference of extractants constituents in testing samples causes a complex pattern of signals reducing the performances of the characterization and deconvolution of the data files. A treatment of the extract by NADES removal, aimed to improve the quality of signals and analysis, could be necessary. In view of green methodology for the extraction of target compounds, NADES can be recycled. Normally, distillation is the current method used for the recycling of organic solvents for the easy recovery of target compounds. The evaporation of NADES extractants is a difficult task because of low vapor pressure and peculiar physiochemical properties causing a problem for industrial applications [54,55]. The possibilities for target compounds recovery and NADES recycling include liquid–liquid extraction (LLE) using another solvent, solid–liquid extraction using a macroporous resin, and the addition of antisolvents [56,57,58]. NADES exhibit unusual solvation properties with protic solvents (methanol, ethanol, or water) due to hydrogen bonding, whereas aprotic ones (toluene, hexane, ethyl acetate, acetonitrile, diethyl ether, etc.) are immiscible. The selective solubility of NADES could be a crucial point for liquid–liquid extraction. Some examples were given by Liu et al., 2016 who used different organic solvents mixtures for the separation and purification of different compounds: rutin was recovered by ethyl acetate/butanediol/H2O (6:4:10, *v*/*v*), quercetin and daidzein by hexane/ethyl acetate/MeOH/H2O (3:7:5:5, *v*/*v*), and kaempferol was isolated by hexane/ethyl acetate/MeOH/H2O (4:6:5:5, *v*/*v*). They obtained a recovery of 95.7, 94.6, 97.0, and 96.7% for rutin, quercetin, kaempferol, and daidzein respectively [59]. Another purification technique is Solid-Phase Extraction (SPE) by using adsorption cartridges. In SPE purification, less polar analytes are adsorbed onto the resins and then it is necessary to wash the NADES out with water followed by the elution of the target analytes with an alcoholic solvent (e.g., ethanol) [56]. Instances of flavonoid isolation through SPE were performed by Wang and Wang, 2019. After optimized ethylene glycol–choline chloride–UAE extraction, authors tested the adsorption and desorption of flavonoids from Safflower on five types of macroporous resins for flavonoids isolation, finding that the highest adsorption capacity was produced by NKA-2 followed by S-8 [60]. On the contrary, Zhuang et al. 2017 selected LX-38 as the best macroporous resin for the recovery of the flavonoids from the DES with a satisfactory yield of 98.92% [61]. Similarly, Bi et al.,2020 performed the flavonoid separation by NADES-MAE followed by direct macroporous resin adsorption and desorption process. Using the optimized parameters, the direct separation of baicalin, wogonoside, baicalein, and wogonin enriched fractions obtained from NADES extraction solution was efficiently achieved using macroporous resin ME-2 with recovery yields around 80–85% [62]. Panìc et al., 2019 proved that the presence of >50% (*v*/*v*) of water ruptures the NADES structure, a simple dilution allowed an anthocyanin recovery of 99.46%, and a highly efficient solvent recycling (yield 96.8%) when authors diluted the grape-pomace extract in 80% of water. The cleanness of the NADES after recycling was proved using NMR spectrometry and no significant differences between freshly synthesized and recycled NADES in 1H NMR spectra were observed. These results implicate that NADES structures should be disturbed prior to recycling via dilution with >50% amount of water (*v*/*v*) to release anthocyanins and make them available for better adsorption on macroporous resins [63].

### 6.2. Extraction of Bioactive Compounds from Natural Sources

Sustainable and highly efficient extraction of bioactive natural compounds from natural sources is considered an important task for the development of nutraceuticals and food supplements. The biodegradability, low toxicity, and adjustable solvent properties of NADES along with their extraordinary solubilizing power for natural products of diverse polarity, accent the interest in their use for the extraction of bioactive compounds from natural sources. Coumaric acid derivatives, flavonoids, anthocyanins, and resveratrol are only a few metabolites ranked in the several classes of phenolics contained in functional foods exerting antioxidant properties. Several studies reported a higher efficiency for the extraction of phenolics from different matrices as reported in Table 2. Luoxuan Lin in 2022 optimized the extraction of phenolic acids from orange peels by using different ratios of choline chloride, glucose, and water. The DPPH assay was used to evaluate the antioxidant properties of extracted phenolic acids. The optimized condition for improved antioxidant activity was obtained at a molar ratio of 5:2:5 (ChCl/D-(+)-glucose/H2O) [64]. Zannou et al., 2022 studied the incidence of a ChCl/acetic acid system on the recovery of total phenolics from bitter melon (*Momordica charantia*) monitored by TPC, TFC, DPPH, and FRAP assays. The tests showed a higher amount of phenolics in agreement with the optimal antioxidant activity for the extracts obtained in the conditions reported in Table 2. The amount of gallic acid, vanillic acid, epicatechin, and quercetin-3-O-glucoside, evaluated by chromatographic analysis, was higher in the considered NADES than in other used extracts [65]. Guo et al., 2019 selected a choline chloride/citric acid/glucose (1:1:1) NADES system for a high-yield extraction of anthocyanins from mulberry by using innovative high-speed homogenization (HSH) and cavitation burst extraction (CBE) [66]. Zengin et al., 2022 performed a comparative LC-ESI-QTOF-MS study of the bioactive compound profiles of *Cytinus hypocistis* extracted by three different choline chloride, proline, and xylitol-based NADES systems compared with conventional solvents, as detailed in Table 2. The correlation of prepared extracts with bioactivity was performed in terms of phenolic and flavonoid content yield along with specific enzyme inhibition. TPC, TFC, DPPH, and ABTS assays were performed for the evaluation of antioxidant activity while cholinesterase (AChE and BChE), tyrosinase, and glucosidase inhibitory assays were performed for specific target interaction. The study highlighted the improved ability of NADES systems in bioactive content yield and enzyme inhibitory bioactivity compared to traditional solvents [67]. Vieira et al., 2022 focused on the advantage of the NADES characterized by different polarity and viscosity to form a biphasic system for the combined extraction of polar rosmarinic acid and non-polar carnosic acid and carnosol compounds from Rosmarinus officinalis [68]. The NADES selective extraction of bioactive molecules from soursop leaves (*Annona muricata* L.) was investigated by Castro Leal et al., 2022. HPLC-DAD monitoring of the main antioxidants rutin and catechin showed the extraction capacity of all the selected NADES systems revealing those composed of glycerol and xylitol as hydrogen bond donors presented the higher extraction indexes [69]. Chen et al., 2022, investigated for the first time NADES for extracting agents of pectins from mango peels. Two novel green solvents were screened betaine-citric acid (Bet-CA) and choline chloride-malic acid (ChCl-MaA) coupled with the USAE preparation technique. The NADES-extracted pectins resulted higher in extraction yield, larger molecular weight, and particle size than HCl-extracted pectins characterized by Fourier transform infrared spectra (FT-IR) and thermal analysis [70]. Bajkacz et al., 2017 developed NADES extraction procedure and UHPLC-UV monitoring for the determination of isoflavones in soy-containing food samples. The optimized conditions for isoflavones extractions were selected with choline chloride/citric acid (1:1) with a 30% water content in NADES [39].

The versatile polarity modulation of NADES systems allows the recovery of a bioactive molecule with non-polar behavior from natural matrices. Several lipid-soluble compounds such as curcumin and lycopene are considered for industrial importance for both their coloring attributes but also for their bioactive properties such as antioxidant, anticancer, and immunomodulatory [71]. Alioui et al., 2022, performed different solubility tests of the non-polar curcumin by using different NADES systems with the aim to demonstrate that the studied NADESs had critical structural features in H-bonding and curcumin solubilization capabilities. Curcumin resulted more soluble in the choline chloride/glycine system than the other NADES examined [72]. The constantly increasing lycopene demand for food, pharmaceutical, and cosmeceutical applications, has led the scientific community to explore alternative approaches to extract it from natural sources. Different hydrophobic natural deep eutectic solvents (HNADES) based on terpenes (i.e., menthol and thymol) and fatty acids (i.e., decanoic acid and dodecanoic acid) were prepared at different molar ratios. Spectrophotometry and RP-HPLC-DAD were used to monitor the process efficiency of extracting lycopene from tomatoes. thymol/lauric acid system (2:1) was the optimized NADES for the selective recovery of the non-polar compound [73]. Four kinds of NADES were tested by Hong et al., 2022 for the extraction of solanesol from tobacco leaves and compared with conventional organic solvents. Solanesol was recovered from the NADES extract by microextraction with the addition of ethyl acetate and finally analyzed using high-performance liquid chromatography (HPLC). The chlorine chloride/urea NADES was selected for the improved extraction efficiency of solanesol, with a water content of 5% (*w*:*w*) [74]. Essential oils are characterized by their aroma and significant pharmacological effects as analgesic, anti-inflammatory, and cerebral activities with nutraceutical and cosmetical interest [75]. Fun and Li., 2022 studied the different NADES for essential oil extraction from *Angelica sinensis* radix. It was found that choline chloride and citric acid were the optimized combination for the extraction of essential oil. The higher composition of the essential oils extracted by the ChCl/citric acid system was analyzed by gas chromatography-mass spectrometry (GC-MS), highlighting the occurrence of ligustilide as the main component of *Angelica sinensis* essential oil [41].
foods-12-00056-t002_Table 2Table 2Extraction of target compounds from food matrices by using NADES and combined extraction techniques.Target CompoundNatural MatrixNADE SystemMolar Ratio *Water/NADES *NADES PreparationsExtraction TechniqueReferenceAnthocyaninsGrape skinCitric acid/ D-(+)-maltose4:13:7Freeze-drying methodUSAE[57]AnthocyaninsMulberryChCl /citric acid/glucose1:1:1-Heating and stirringHSH andCBE[66]Bioactive compounds*Cytinus hypocistis*Proline/xylitol5:1-HeatingUSAE[67]Bioactive compounds*Annona muricata*ChCl/ xylitol1:1-
USAE[69]Essential oils*Angelica sinensis* radixChCl/citric acid1:1-Heating and stirringHydrodistillation microwaves[41]PectinsMango peelBetaine/citric acidcholine chloride/malic acid1:11:1--Heating and stirringUSAE[70]Phenolic acidsOrange peelChCl/D-(+)-glucose/H_2_O5:2:5
MixingUSAE[64]Phenolic compoundsBitter melonChCl/acetic acid1:4.351:5HeatingUSAE[65]Phenolic compounds*Moringa oleifera*Proline/glycine1:11:3Heating and stirringUSAE[76]Phenolic compounds*Olea europaea*H_2_O/ChCl/fructose5:5:2
MixingUSAE[77]Polyphenols*Sambucus nigra*Glycine/lactic acid1:115%HeatingUSAE[78]Flavonoids*Radix scutellariae*Proline/glycine1:11:3StirringUSAE[79]Tannins*Osmansthus fragrans*Lactic acid/glucose1:14:1Heating and stirringMAE[80]CaffeineChinese dark teaChCl/lactic acid1:11:3HeatingSLE[81](+)-catechinGrape skinChCl/Oxalic acid1:11:4Heating and stirringUSAE[82]CurcuminStandard solubility testsChCl/glycine1:1-Heating and stirringUSAE[72]IsoflavonesSoy beansChC/citric acid1:130%Stirring and heatingUSAE[39]LycopeneTomato fruitThymol/lauric acid2:1-Heating and stirringStirring[73]SolenesolTobacco leavesChCl/urea1:25%Heating and stirringUSAE[74]Tryptanthrin, indirubin, and indigo*Baphicacanthus cusia*Lactic acid/L-menthol1:1-USAEUSAE[83]Ursolic acid*Cynomorium songaricum Rupr*.ChCl/ D-(+)-glucose1:1-HeatingUSAE-ATPS[84]Rosmarinic acid, carnosol, carnosic acid*Rosmarinus officinalis*Lactic acid-glucose/menthol-lauric acid (biphasic system)5:1/2:1-Heating and stirringUSAE[68]* (*v*/*v*) molar ratio. “-“ not provided.


### 6.3. Extraction of Bioactive Compounds from Agricultural Food by-Products

The awareness for the optimization of natural resources and the environmentally sound management of chemicals and all wastes throughout their life cycle is growing [49]. A big amount of food waste produced (about 1.3 billion tons of food waste per year) could provide a useful and inexpensive source to obtain high-value compounds [85]. Moreover, the use of food by-products as a source of bioactive compounds is a good opportunity to implement a circular economy and reduce the continuous increase in the production of organic wastes [86] and its consequent environmental problem. In particular, the use of by-products as a source of biomass for cosmetical, nutraceutical products, or biofertilizers can create profit and reduce the environmental impact and the costs of their disposal.

The olive oil industry produces large biomass of not-edible products, olive pomace is at least 60–65% of the total weight of olives. As well as the whole olive, pomace is considered a source of phenolic bioactive compounds. The NADES selective extraction of phenolics from this complex matrix was performed by Chanioti et Tzia 2018 (Table 3). The NADESs obtained by choline chloride and organic acid HBD, as citric acid and lactic acid, were selected as the most promising extractants and more effective in the olive pomace phenolic compounds extraction compared to conventional solvents. The extraction technique by homogenization possessed the highest total phenolic content (TPC) and antioxidant activity of the other extracts [87]. Nelus Mayer et al., 2022 also focused their attention on the optimization of NADE extraction of phenolics on pomace olive. A comparative study between different NADES and conventional solvents obtained by ultrasound-assisted extraction technique was performed by HPLC monitoring of the hydroxytyrosol and luteolin concentration, and the determination of total phenolic content, anthocyanin content, and antioxidant capacity in samples from different cultivars and harvest seasons. Lactic acid/glucose was selected as the optimized extractant of bioactive phenolics using an ultrasound technique [88]. Olive leaves are another important waste product for the alimentary production of oil. Zurob et al., 2020, screened the extraction efficiencies of eight sugar-based and organic acid-based NADES for the recovery of hydroxytyrosol. The HPLC-UV experimental results showed that the hydroxytyrosol amount is higher in a citric acid/glycine/water (2:1:1) system than in water used as a conventional solvent for the extraction of polyphenols [89].

The recovery of anthocyanins by using NADES from grape pomace was studied by Panić et al., 2019. A comparative study of several NADES for the optimization of composition, physiochemical characteristics, anthocyanins extraction power, and solvent price was investigated and performed on a larger scale-up extraction process with NADES recycling. The highest content of all the anthocyanins, evaluated by the sum of all HPLC signals, was attributed to the ChCl/citric acid and ChCl/proline/malic acid systems by using an ultrasound microwave-assisted extraction technique [63]. In the recent work of Lanjekar et al., 2022, eight different NADES were prepared using components such as choline chloride, carboxylic acids, sugars, and alcohols to assess the extraction efficiency from mango peel by-products. The lactic acid/glucose system exerted the highest total phenol and flavonoid content [90].
foods-12-00056-t003_Table 3Table 3Extraction of target compounds from food wastes by using NADES and combined extraction techniques.Target CompoundNatural MatrixNADE SystemMolar Ratio *Water/DES *Extraction TechniqueReferencePhenolic compoundsOlive pomaceChCl /citric acid1:220%Homogenization[87]Phenolic compoundsOlive pomaceLactic acid/glucose/water5:1:9.3
USAE[88]Phenolic compoundsHazelnut skinChCl /lactic acid1:235%USAE[91]Phenolic compoundsCocoa beansBetaine/glucose5:2
USAE[92]Phenolic compoundsWaste mango peellactic acid/glucose5:120%USAE[90]AnthocyaninsGrape pomaceChCl /citric acidChCl /proline/ malic acid2:11:1:130%25%USAE/MAE[63]AnthocyaninsSour cherry pomaceChCl/malic acid1:120%MAE[33]AnthocyaninsBlueberry peelChCl/malic acidChCl/citric acid1.5:12:122%MAE[93]HydroxytyrosolOlive leavesCitric acid/glycine/water2:1:1
USAE[89]* *v*/*v* ratio.


### 6.4. Food Safety

While natural phenolic compounds are involved in plant biological processes such as growth, reproduction, and defense mechanisms, synthetic phenolic compounds (i.e., phenol, o-cresol, and 2-chlorophenol) are precursors of nylon, and detergents used to produce polyphenoxy polymers, fertilizers, paints, or explosives. The World Health Organization, WHO, lists some of these phenolic compounds as priority pollutants due to their toxicity, so it is necessary to remove them, to a concentration below the legal limit before wastewater discharge [94]. G. Sas et al., 2019, texted six new NADES based on organic acids (dodecanoic acid, decanoic acid, octanoic acid) and menthol or thymol, as extraction solvents for these pollutant compounds. The UV–Vis spectrophotometer was used to quantify the phenolic compound concentrations in the aqueous phase after extraction. Menthol/octanoic acid and menthol/decanoic acid (both with molar ratio 1:1) presented the best results for the extraction of phenolic pollutants from water obtaining extraction yields upper than 80% for 2-chlorophenol and o-cresol compounds (Table 4) [94]. The analysis of bisphenols and alkylphenols in functional beverages, kombucha, and water kefir was investigated by Baute-Pérez et al., 2022. The main objective of their paper was to evaluate the presence of 13 alkylphenols, bisphenols, and alkylphenol ethoxylates in bottled water, kombucha, and water kefir by using NADES extractants, based on monoterpenes and fatty acids, and detected by a new vortex-assisted (VA)-DLLME-ultra-high-performance liquid chromatography-mass spectrometry (UHPLC-MS) as defined in Table 4. Only TOP1EO was detected in bottled waters, whereas no residues appeared in kombuchas and water kefirs. The analytical performance of the method and the obtained results demonstrated not only the applicability of the method but also that the consumption of these matrices is safe [95]. Soltani and Sereshti 2022 developed a novel green analytical procedure based on green deep eutectic solvents (DES) followed by GC-MS to analyze pesticides in tea samples. This work aims to develop a green QuEChERS approach modified based on the use of green deep eutectic solvents (DES) for two basic steps: (1) the extraction of pesticides from tea samples, and (2) the sample clean-up prior to GC-MS analysis. Among five tested hydrophilic DESs, ChCl/polyethylene glycol (1:4 M ratio) DES was selected and used as a green extractant. For the clean-up of the tea extract, magnetic 3D-graphene aerogel (3DG-Fe_3_O_4_) was functionalized with a choline chloride/urea NADES system used as a new sorbent in the in-syringe dispersive μSPE method. A comparative GC-MS study between 3DGA-Fe_3_O_4_ and the modified 3DGA-Fe_3_O_4_/NADES obtained samples showed that functionalization with natural NADES resulted in higher extraction efficiencies (1.4–1.7 fold) [96]. As discussed before, by-products generated in agro-food industries are a valuable source of numerous bioactive compounds that can be used for the prevention and treatment of several diseases or as nutritional ingredients in functional foods, for the preparation of cosmetics. The determination of pollutants in citrus and olive waste products is a crucial point for the safety of products with human use. An analytical method for the detection of pollutants in olive and citrus by-products was studied by Socas-Rodrìguez et al., 2022. A comparative GC-MS determination of pesticides in several betaine-based NADESs used as extractants for the ultrasound-assisted extraction (UAE) from olive and citrus by-products. A group of 12 pesticides (4 organophosphate pesticides, 3 organochlorinated, 1 chlorotriazine, 1 tiadiazine, 1 strobilurin, 1 pyrazole, and 1 pyrethroid), commonly used in olive and citrus crops and during storage processes, was monitored to evaluate the safety of valorized by-products with neuroprotection potential against neurodegenerative diseases such as Alzheimer’s disease. The combination of betaine and 1,2-propylene glycol (in a molar ratio 1:4) provided the highest extraction efficiencies for the largest number of screened compounds [97]. Sereshti et al., 2022 developed a green analytical procedure to analyze pesticides in water samples. A new green alternative DLLME based on two novel NaDESs, thymol/myristyl alcohol (2:1) and alanine/kojic acid/water (1:2:5) coupled to GC-µECD method was conducted to determine 16 multiclass pesticides in water samples. The results showed that no detectable target analytes were observed in seawater and drinking water tap samples. However, bromopropylate was detected in river water and groundwater; β-endosulfan was found in groundwater; and fenpropathrin was seen in groundwater [98].

In addition to organic pollutants, the presence of metals in foods is currently an environmental and health problem. Cadmium (Cd) is a common and toxic ecological pollutant, which originates from swift industrial operations, the overuse of fertilizers, composts and sewage sediments, and municipal activities [99,100]. In the work of Shamsipur et al., 2022, a comparison of the analytical performances achieved from the ultrasound-vortex-assisted liquid-phase microextraction technique and previously reported techniques for the extraction and determination of Cd (II) ions was detected. A hydrophobic natural deep eutectic solvent composed of salicylic acid and l-menthol (at a 1:4 molar proportion) was used as both the complexing agent and the extraction solvent of the metal. The performed extraction technique was favorably applied to the separation and determination of ultra-traces of cadmium in several food and water samples. Among the used determination tools, graphite furnace atomic absorption spectrometry (GFAAS) was established as a highly efficient technique for the determination of the concentration of various metal ion species [101].

The determination of mycotoxins also interferes with the food safety guarantee. Zearalenone is one of the most common mycotoxins produced by Fusarium fungi [102]. Pochivalov et al., 2022, studied extraction systems applied to the mycotoxin (zearalenone) determination in cereal samples. A molar ratio between DES components (DL-menthol and 1-hexanol) was investigated in the range from 1:2 to 3:1. The studied extraction system was successfully applied to zearalenone separation from cereal samples followed by its sensitive determination. The stability of the deep eutectic solvent in an acetonitrile–water mixture was studied using gas chromatography-flame ionization detection and the Karl Fisher method. The obtained results are useful for future developments in this area and are aimed at drawing the attention of the researchers to the stability issues with hydrophobic DESs in polar solvent–water mixtures [103]. 5-hydroxymethylfurfural (HMF) is a breakdown product and is practically absent in fresh food, but it is naturally generated in sugar-containing food during heat treatments such as drying or cooking because of the Maillard reaction. It is particularly toxic to human health, but it has been regarded as an essential building block for synthesizing chemicals and biofuels. Until now, the efficient preparation of HMF from biomass is still very challenging, and it needs to overcome several steps of catalytic reactions. The work recently published by Zuo et al., 2022, aimed to provide a simple, green, and cheap route for HMF production from raw biomass and suggest a new idea for biorefinery technology. Glucose was initially selected as a model feedstock to optimize the reaction conditions in A-NADES, as well as to uncover the catalytic mechanism of the glucose isomerization and dehydration steps, with the low amount of metal chlorides tested as the catalysts. Subsequently, starch and food waste were applied as raw feedstocks to test the comprehensive utilization efficiency of the NADES system and study its benefits in relieving environmental pollution. Furthermore, they also provide a food waste disposal procedure with considerable environmental relevance owing to the immense volumes of food waste generated globally. We demonstrated that aqueous natural deep eutectic solvent ChCl/glucose/water (2:1:1) showed remarkable performance for converting glucose, starch, and food waste into 5-hydroxymethylfurfural (HMF) combined with SnCl4 catalyst. More importantly, food wastes such as rice waste and bread waste were effectively converted into HMF with a high yield of 61.3 and 54.5%, respectively, indicating the NADES system has a good potential for food waste. High HMF yields of 64.3, 64.0, 61.3, and 54.5% were obtained from glucose, starch, rice waste, and bread waste at 130 °C in the NADES/MIBK (methyl isobutyl ketone) biphasic system, respectively [104].

The selective extraction of allergens from food is required to ensure the health care of consumers with alimentary allergies and intolerances. Gluten is the main allergen commonly diffused in everyday products based on flavors of cereals such as grains such as wheat, rye, barley, and oats. In food processing, the removal of the protein occurring in the endosperm of cereals is obtained by extraction with an ethanol–water solution. The current approach could produce oxidated forms and structure degradation of the protein causing the incomplete extraction of modified peptides from the food. Lores et al. 2017 performed an ultrasound-assisted extraction in combination with the use of fructose/citric acid NADES [105]. After the extraction step, ELISA tests were performed to monitor the production of, the solubilized proteins maintained their structures well without significant changes. An inverse relationship was found between the viscosity and solubilization ability of NADES, and a solvent with a viscosity close to that of water appeared to be most effective for gluten extraction. Additionally, the presence of citric acid in the NADES can help prevent proteins from oxidation, thus eliminating the need for additional reducing agents such as β-mercaptoethanol, normally required for ethanol–water extraction.
foods-12-00056-t004_Table 4Table 4Food safety evaluation using NADES and combined extraction techniques.Target CompoundMatrixNADE SystemMolar Ratio *Extraction TechniqueRef.2-chlorophenol, O-cresolWastewaterMenthol/octanoic acid Menthol/decanoic acid1:1Liquid–liquid extraction[94]Alkylphenols, bisphenols, and alkylphenol ethoxylatesFunctional beverageBottled waterMenthol/octanoic acidMenthol/decanoic acidThymol/octanoic acidThymol/decanoic acidMenthol/octanoic acidMenthol/octanoic acid1:11:11:11:11:22:1Vortex-assisted dispersive liquid–liquid microextraction[95]PesticidesGreen and black teasChloride/polyethylene glycolChCl/urea 1:4-NADES functionalized 3D-graphene aerogel (3DG-Fe_3_O_4_) aerogel[96]PesticidesCitrus and olive byproductsBetaine/1,2-propylene glycol1:4USAE[97]PesticidesWaterThymol/myristyl alcoholAlanine/kojic acid/water2:11:2:5Dispersive liquid–liquid microextraction[98]CadmiumWater andfood samplesSalicylic acid/l-menthol1:4Ultrasound-vortex-assisted dispersive liquid–liquid microextraction[101]ZearalenoneCerealsMenthol/1-hexanol2:1Dispersive liquid–liquid microextraction[103]5-hydroxymethylfurfuralGlucose, starch, and food wastesChcl/glucose/water2:1:1Heating and stirring[104]* *v*/*v* ratio. “-“ not provided.

## 7. Conclusions and Future Prospective

The selection of the optimum extraction solvent is a crucial step to achieving a good extraction efficiency of target analytes, and at the same time minimizing the interfering compounds, reducing the environmental impact. The growing awareness to decrease pollution and energy dispersion suggests the replacement of organic solvents with alternative systems characterized by lower environmental impact and human toxicity. In this view, the green solvents NADES has been increasingly used as sustainable solvents in several applications of food analysis in both the research community and the food industry. The current review collects several studies about the physicochemical properties of NADES, their preparation methods, evaluation of toxicity, and their use in combination with green extraction techniques for providing a green chemistry approach to several aspects of food analysis. In addition, this review also provides an update on a series of new applications of NADES in the extraction of bioactive compounds from foods and their by-products and in the determination of contaminants in foods. However, a deeper understanding about the safety of NADES systems on human health and their chemical stability is also another aspect that deserves to be further researched.

## Figures and Tables

**Figure 1 foods-12-00056-f001:**
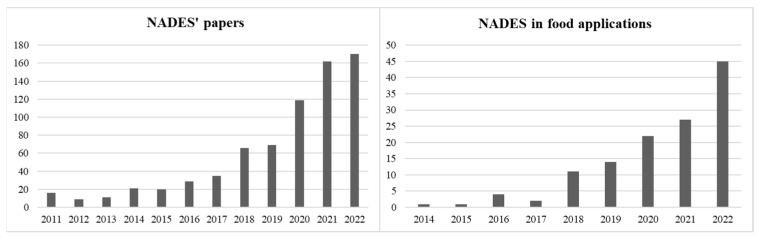
The number of papers published (source Scopus) since 2011 focusing on NADES (panel left) and NADES in food applications since 2014 (panel right).

**Figure 2 foods-12-00056-f002:**
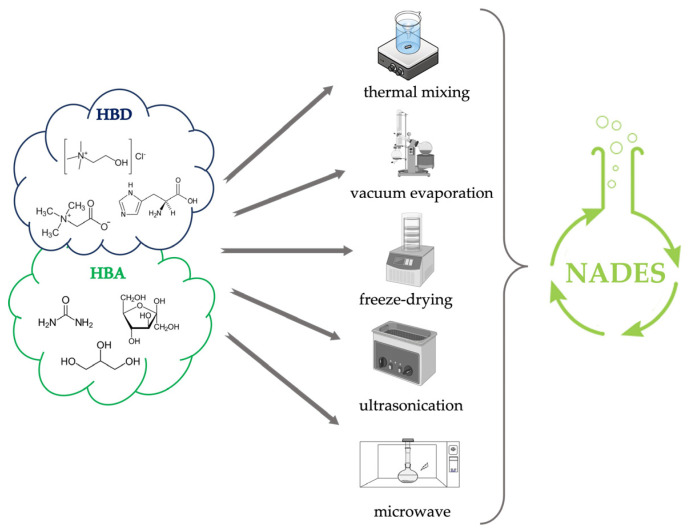
Graphical representation of NADES preparation techniques.

**Table 1 foods-12-00056-t001:** Comparison of different extraction techniques with different mixtures of NADES.

Extraction Technique	Type of NADES	Compounds	Extraction Yield	Ref.
Heating and stirring	ChCl:MalA	Phenols	3.2 mg/g ^a^	[33]
	ChCl:Ur	Phenols	2.7 mg/g ^a^	
	ChCl:Fru	Phenols	1.80 mg/g ^a^	
USAE	ChCl:MalA	Phenols	2.5 mg/g ^a^	
USAE	ChCl:glycerol	Phenols	5.6 mg/g ^b^	[2]
MAE	ChCl:MalA	Phenols	3.0 mg/g ^a^	[33]
PHWE	ChCl:Ox	Anthocyanins	170.0 ± 6.5 mg/g ^c^	[40]
	ChCl:La	Anthocyanins	146.1 ± 63.9 mg/g ^c^	
	ChCl:Fru	Anthocyanins	78.5 ± 6.5 mg/g ^c^	
	ChCl:EtOH	Anthocyanins	93.7 ± 8.24 mg/g ^c^	
	ChCl:Pro	Anthocyanins	145.5 ± 4.9 mg/g ^c^	
	ChCl:Ur	Anthocyanins	101.4 ± 27.9 mg/g ^c^	
	CaMa	Anthocyanins	39.2 ± 9.8 mg/g ^c^	
	CaFru	Anthocyanins	47.4 ± 1.6 mg/g ^c^	

^a^ mg_total phenol_/gEXT, ^b^ mg_gallic acid equivalent_/gEXT, ^c^ mg_malvidin-3-glucoside equivalent_/gEXT.

## Data Availability

Data is contained within the article.

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
