# Peer review of "Natural Deep Eutectic Solvents (NADESs) Combined with Sustainable Extraction Techniques: A Review of the Green Chemistry Approach in Food Analysis"

_foods, 2022, doi:10.3390/foods12010056_

Round 1

Reviewer 1 Report

This review is devoted to the topic of Natural Deep Eutectic Solvents and their application in food science. The article is relevant and interesting. A good structure, the presence of generalizing tables are also the advantages of this work. I recommend making the following improvements:

1. It is necessary to point out more clearly the advantages of this review over a large number of reviews on eutectic solvents.

2. It would be nice to add graphic material to this article.

3. Please cite: 10.1016/j.molliq.2022.119859.

4. "Conclusions and future prospective" can be described in more detail.

Author Response

Dear Reviewer, we would like to thank you for your opinion on our paper, which encourages us to correct some criticisms.

attached you can find a detailed response to your comments

Reviewer 2 Report

I reviewed the review manuscript entitled, Natural Deep Eutectic Solvents (NADESs) combined with sustainable extraction techniques. A review of the green chemistry approach in food analysis. The review is complete and contributes to the field. In my opinion, this manuscript can be addressing below points

Abstract

Authors should revise the abstract based on what are the review findings? What authors want to conclude and recommend from this review? Authors just focused on the background of the study and review objectives.

Figure 1. improved the quality of the Figure and provide the source. Is it from the web of science or something else? How has the information been retrieved?

Lines 71 to 74: seems like review objectives and should moved to the end of Introduction

Line 96: The The intermolecular….?

Line 98: place the full stop after the reference

Section 3. Please PROVIDE A FLOW CHART representing the preparation of NADES

Section 4: Please provide a Table containing extraction techniques that use green solvents including which compounds are extracted and what is the yield of extracted compound?

Section 5. Toxicity and sustainability: Please provide a table containing the toxicity levels of green solvent. How toxic is it? Levels?

Section 6 is complete and appropriate

All of the references are not according to the format. Please revise it.

Author Response

We would like to thank the reviewer for all the useful comments on manuscript, which highlighting some criticisms and surely will increase the quality of manuscript.
